# Biodegradable Temporising Matrix for Lower Limb Reconstruction following the Resection of Giant Marjolin's Ulcer

Samuel MacDiarmid * and Daniel Butler *

Department of Plastic Surgery, Tauranga Hospital, Tauranga 3112, New Zealand
* Correspondence: samuelmacdiarmid@gmail.com (S.M.); daniel.butler@bopdhb.govt.nz (D.B.)

**Abstract:** NovoSorb® Biodegradable Temporising Matrix (BTM) is a synthetic matrix used as an adjunct in the reconstruction of certain complex wounds. We present a gentleman who sustained severe full-thickness lower limb burns as a child which were treated with split-thickness skin grafts. In later life, he went on to develop bilateral non-healing ulcers, resulting in a left above-knee amputation and a giant circumferential right lower limb squamous cell carcinoma (SCC) encompassing the majority of the lower leg. Surgical resection and salvage of the single remaining limb was achieved with the successful application of a BTM. BTM has proven to be successful in reconstructing a small number of SCC wounds; however, to the best of our knowledge, we are the first authors to test its application in the reconstruction of a circumferential defect associated with a giant lower limb Marjolin's ulcer.

**Keywords:** biodegradable temporising matrix; squamous cell carcinoma; Marjolin's ulcer; lower limb; limb reconstruction



## 1. Introduction

Marjolin's ulcer describes the development of cutaneous malignancy within an area of previously damaged or inflamed skin [1]. Histology predominately demonstrates squamous cell carcinoma (SCC), but basal cell carcinoma, melanoma, and sarcoma have also been described [2]. These tumours most commonly occur within old thermal burn scars, but can be associated with a wide range of other non-healing wounds [3]. Treatment for localised disease is variable but usually includes Mohs surgery, wide local excision, or amputation [1]. The disease is typically considered high risk, and despite a lack of evidence, adjuvant radiotherapy is frequently offered in an attempt to optimise local control, avoid recurrence, and minimise the risk of metastasis.

NovoSorb® Biodegradable Temporising Matrix (BTM; PolyNovo Biomaterials Pty Ltd., Port Melbourne, Victoria, Australia) is an implantable bilayered synthetic matrix which is intended for the reconstruction of complex wounds [4]. It comprises a 2 mm biodegradable polyurethane foam matrix covered by a non-biodegradable fenestrated sealing membrane which is applied in a two-step procedure [4,5]. The first stage involves implantation with sutures or staples and negative-pressure wound therapy (NPWT). The second stage is performed when adequate wound integration is achieved and entails delamination of the sealing membrane along with split-thickness skin graft (STSG) application to the neodermis. This results in the formation of a robust dermal substitute which adds volume to wounds, improves graft blood supply, and facilitates graft take over both tendon and bone [4]. A lack of xenogenic material enables BTM to overcome the risk of interspecies immune rejection, disease transmission, and cultural obstacles associated with animal-derived alternatives [6].

Here, we present a left lower limb amputee with a circumferential Marjolin's ulcer encompassing 20 cm of the right lower leg. This lesion was successfully excised and the defect was reconstructed with BTM and STSG, negating the need for free tissue transfer or

a second above-knee amputation. This approach has allowed us to achieve an excellent balance between function, aesthetic, resource and initial oncological outcome.

## 2. Case

Our patient is a 76-year-old male who sustained bilateral full-thickness lower limb burns after falling into a geothermal pool at 12 years of age. Both of his legs were salvaged at the time and treated with extensive STSGs. He was left with gross soft tissue lower limb deformity, but was still able to ambulate independently. In later years, he developed bilateral non-healing ulcers over the previous graft sites requiring multiple debridement procedures. The ulcer on the left side failed to improve with conservative management and began to significantly impair the patient's quality of life. Subsequent left above-knee amputation was performed by a provincial general surgical team in 2018. Pre-operative mapping biopsies showed inflammatory changes with squamous hyperplasia, but no evidence of malignancy. His single right leg presented as a weight-bearing, sensate stump which supported transfer between bed and wheelchair. It was essential in maintaining his mobility and independence. Co-morbidities included hypertension, dyslipidaemia, and impaired glucose tolerance.

Ulcer surveillance on the right lower limb was performed by community district nurses and the patient's family doctor. He was referred to our service due to deterioration in the right lower limb wound which had progressed to involve the majority of the lower leg in a circumferential nature (Figure 1A). Subsequent punch biopsies of the ulcer confirmed the presence of widespread and well-differentiated SCC. Urgent outpatient CT and MRI demonstrated extension down to the muscle fascia, but with no evidence of invasion into the underlying muscle, periosteum, or bone. Staging CT also identified enlarged ilioinguinal lymph nodes on the right-hand side. PET CT and USS-guided core biopsies deemed these nodes to be reactive rather than malignant.

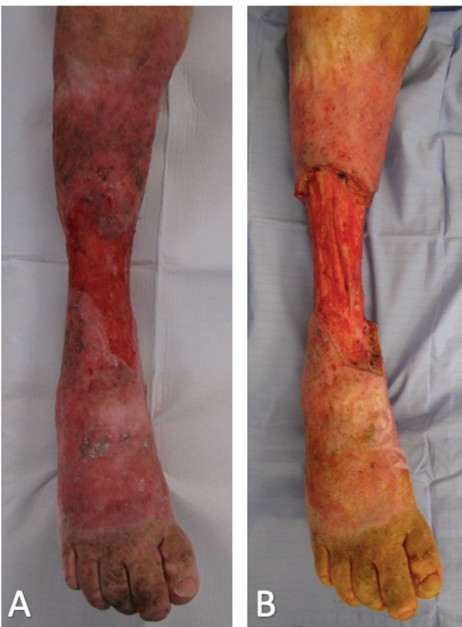

**Figure 1.** Right leg giant squamous cell carcinoma: (**A**) Pre-resection (**B**) Post-resection.

Surgical options included amputation or excision with lower limb reconstruction. Consensus from our multidisciplinary team was that limb salvage could be achieved but adjuvant radiotherapy may be required depending on the histological features of the SCC. Reconstructive options included free flap repair, isolated STSG, or the application of BTM with second-stage STSG. Reconstruction with BTM and STSG represented an optimal solution that had the potential to overcome the major disadvantages associated with free

flap repair and isolated STSG. All surgical options were discussed with the patient, but given his desire to maintain independence, he opted for limb-conserving surgery. He was advised that adjuvant radiotherapy would be offered within 8 weeks of surgery should his post-operative histology demonstrate any unfavourable features (poor differentiation, perineural/lymphovascular invasion, or fascial involvement). Both written and verbal consent were obtained prior to surgery.

Excision was performed in the subfascial plane, sacrificing the long and short saphenous vein, superficial peroneal nerve, sural nerve and saphenous nerve. This resulted in a 22 cm long circumferential defect exposing the muscles and paratenon of the right leg (Figure 1B). The BTM was then cut to size, inset with surgical staples (Figure 2A), and covered with a non-adherent permeable silicone dressing and circumferential NPWT bandage.

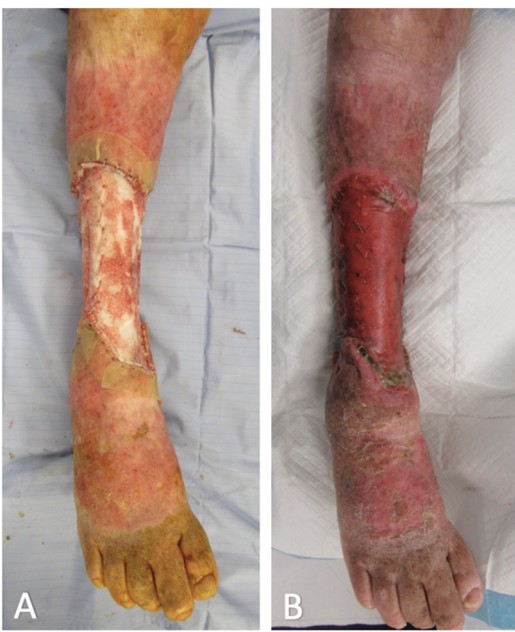

**Figure 2.** Right leg wound with Biodegradable Temporising Matrix in situ: (**A**) Initial implantation (**B**) 5 weeks post-implantation.

Delamination of the BTM sealing membrane along with STSG reconstruction was performed five weeks after the initial excision. The 0.008-inch STSG was harvested from the right thigh and meshed 1:1.5. BTM integration was noted to be 100% at the time of delamination, with no evidence of infection or haematoma (Figure 2B).

Graft check at one week showed 95% graft take with 5% loss over the distal posterior aspect of the wound, although this region healed well with secondary intention. At five weeks after STSG application, the patient was back to his baseline level of mobility, with a fully healed robust and pliable wound (Figure 3A). Five months after the STSG application, he continued to enjoy full function of the single right limb without any form of wound complication (Figure 3B).

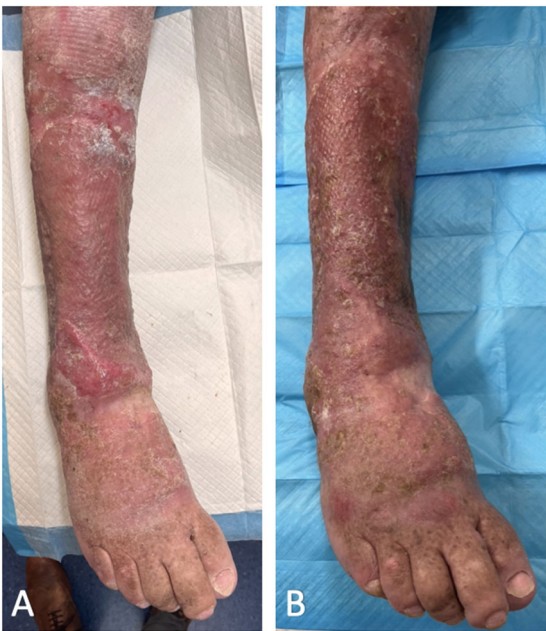

**Figure 3.** Right leg wound: (**A**) 5 weeks after Biodegradable Temporising Matrix delamination and split thickness skin graft; (**B**) 5 months after Biodegradable Temporising Matrix delamination and split thickness skin graft.

Post-operative histology showed a well-differentiated SCC with clear margins (14 mm superiorly, 6.4 mm inferiorly, and 1.8 mm deep with no invasion into the deep fascia), a 3 mm depth of invasion, no extension beyond the reticular dermis, and no vascular or perineural invasion. Given the reassuring histological features, our multidisciplinary team felt that the benefit of radiotherapy was unlikely to outweigh the risk of worsening his foot lymphedema. This was discussed with the patient, who was happy to avoid radiotherapy and continue with indefinite 6-monthly clinical surveillance. The patient is now 9 months post-surgery, with no evidence of recurrence.

### 3. Discussion

The term "Marjolin's ulcer" is typically used to describe a rare and aggressive SCC which arises from chronic wounds [3,7]. The disease most commonly occurs among burn patients with up to 2% of burn scars undergoing malignant transformation [1,7]. Other causes include venous ulcers, pressure sores, osteomyelitic fistulae, traumatic wounds, vaccination scars, surgical scars, and animal bites [3,7]. Prevalence of the disease is relatively high in areas of socio-economic deprivation, and is thought to be a result of limited health resources, suboptimal acute wound management, poor wound surveillance, and lack of disease awareness [8]. The mean latency period between initial insult and malignant transformation varies significantly between studies, but is estimated to be around 29 years [3,7,8]. Although there are multiple different theories which support malignant degeneration within burn scars, the exact pathophysiology remains unknown [1]. Lesions may develop on any cutaneous surface, but most frequently involve the lower limb followed by the head, upper limb, and torso [8]. The disease carries a poor prognosis with high rates of recurrence and metastasis when compared with other forms of cutaneous SCC [3,7,9]. However, it remains unclear whether this is due to intrinsic tumour behaviour or delayed presentation and diagnosis [9].

The prevention of Marjolin's ulcers centres around good acute burn management, education, and wound surveillance [1]. Currently, there are no high-level evidence-based guidelines for the management of Marjolin's ulcers, and treatment is largely determined by lesion size, anatomical location, staging, surgical experience, and institution capability [1,3,8]. Surgical options for localised disease include Mohs surgery, wide local excision,

or amputation [1]. When tumour resection is a viable option, the majority of authors agree that a 2–4 cm surgical margin should be implemented where able [2]. Mohs surgery has demonstrated improved cancer-free survival rates in select cases, but this technique is expensive, labour-intensive, and often not suitable for more complex tumours [1]. Large soft tissue defects following resection are usually reconstructed with direct STSG or free tissue transfer. Given the high-risk nature of the disease, adjuvant radiotherapy is used in many centres, but its application and results are variable [8]. Furthermore, its efficacy remains unvalidated due to a lack of quality retrospective data [8]. Despite the absence of robust treatment protocols, there is general agreement that all Marjolin's ulcers should be managed through a multidisciplinary approach [8,9].

The surgical treatment of giant lower limb Marjolin's ulcers can be challenging. Historically, options have been limited to amputation or excision with free flap transfer or STSG repair. Amputation tends to provide good margin status, but results in unfavourable functional outcomes. Free flap reconstruction is technically challenging and requires a viable donor site of adequate size, in addition to access to a microscope. Large lower limb defects typically require latissimus dorsi free muscle transfer, but these have the potential to cause weakness in the shoulder girdle, which, in our case, would further impair mobility. Failure rates of free tissue transfer to the lower limb are also higher compared with other regions of the body [10,11]. Although direct STSGs represent a simple reconstructive option, they tend to be less durable in the presence of radiotherapy and contribute to wound contracture [5,12]. Furthermore, graft adherence to underlying structures in the lower limb restricts joint mobility and increases the risk of graft failure [4,5].

BTM with STSG provides an alternative reconstructive option following the excision of large cutaneous malignancy in the lower limb. Its application is less technically demanding than free flap reconstruction and overcomes many of the disadvantages associated with direct STSG application. Although clinical trials are lacking, it is presumed that wounds reconstructed with BTM are more likely to tolerate radiotherapy than wounds reconstructed with STSG alone. This is attributed to the increased vascularity associated with neodermis formation and is an important consideration in any form of onco-plastic surgery. The main disadvantage associated with BTM reconstruction is the need for a two-stage procedure which places further strain on hospital resources and may prolong patient recovery. However, if patient selection is appropriate, the benefits of using this type of reconstruction are likely to far outweigh the cost.

BTM has been successfully evaluated in the treatment of a small number of lower limb tumours; however, this is the first time that its application has been tested in the reconstruction of a circumferential wrapping defect associated with a giant Marjolijn's ulcer [4]. Given our experience, we believe that BTM should be considered as a valuable adjunct for any surgeon dealing with complex forms of this disease.

## 4. Conclusions

This case report has demonstrated successful wound reconstruction using BTM and STSG following the resection of a giant lower limb Marjolin's ulcer. Applications of this technique have facilitated limb salvage, overcome the disadvantages associated with traditional reconstructive approaches, and resulted in a robust wound that would likely tolerate radiotherapy.

**Author Contributions:** Writing—original draft preparation, S.M.; writing—review and editing, S.M. and D.B.; supervision, D.B. All authors have read and agreed to the published version of the manuscript.

**Funding:** This research received no external funding; however, the article processing fee was kindly covered by PolyNovo Biomaterials Pty Ltd.

**Informed Consent Statement:** Written informed consent has been obtained from the patient(s) to publish this paper.

**Conflicts of Interest:** The authors declare no conflict of interest.

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
