# Peer review of "Biodegradable Temporising Matrix for Lower Limb Reconstruction following the Resection of Giant Marjolin’s Ulcer"

_2673-1991, doi:10.3390/ebj3040045_

Round 1

Reviewer 1 Report

Thank you for the opportunity to review this manuscript. I really enjoyed reading it. Well written, easy to follow , interesting with good photographs.

There is a small spelling error of 'ulcer ' in line 27.

BTM has now been used for a wide variety of wounds so I suppose you could say there is a lack of originality here, however readers will be impressed to see how well it reconstructed this difficult wound. In many other centres this would have resulted in amputation. This paper is primarily of interest to plastic surgeons undertaking skin oncology work, although the original causation was as a result of burns.

I have just 2 suggestions for improvement:

1. As this is a burn journal it would be worth a little more discussion around the topic of Marjolin ulcers. How common are they and what is the usual reconstruction offered? are they often associated with lower limb? in recent literature, what is the average lag time after burn injury? I would assume in this case the patient was no longer under Burn team review and was referred back in from dermatology or family doctor? Often by the time these conditions arise the patient is being cared for by other teams who have a lack of awareness of Marjolins. It would be interesting to know how long ago he had his amputation, who undertook that, was Marjolins considered then? Did that specimen have histology done on the ulcers? If available these details could be added to the case history and discussion sections.

2. A little more detail and discussion is needed around the topic of radiotherapy. You mentioned that all options were fully discussed with the patient and presumably he consented to the management plan of limb preservation + radiotherapy. Then the team changed the plan after surgery. Theoretically (although I accept unlikely) he may have opted for amputation to ensure maximum tumour clearance, if he had known the radiotherapy would then not happen. A parallel could be drawn with the discussion with breast cancer patients whether to have mastectomy or breast conservation surgery.

I appreciate he was probably very happy to hear the histology was good and the radiotherapy not needed, but this aspect of the case needs to be explained in more detail, particularly in relation to the consenting at the first stage.

How long has he been followed up for now ? Apologies you may have mentioned that and I may have missed it. Do you plan to closely monitor by any sort of scans or simply clinical examination?

In some way it is a pity that the radiotherapy was not required as this would have been very interesting to see how robust the BTM was and would certainly have given a very novel angle to your report.

I do not know if the journal publishes case reports. If it does, I think this is well worthy of publication and I look forward to seeing it in the journal.

Author Response

Sorry about the delay in reply, I have been away on leave. Thank you for your comments and feedback, all useful and very constructive.

As per your request, I have expanded on Majolin’s ulcers and provided a few more details relevant to the case. I have explored the concept of radiotherapy but struggled to make the link between post-operative radiotherapy in breast cancer vs SCC. This is due to the fact that in our centre radiotherapy and wide local excision for breast cancer is a package deal which is consented for prior to surgery, however, with large SCC’s we will usually advise the patient that radiotherapy is dependent on post-operative histological features and the final decision is made using risk benefit analysis at our multi-disciplinary team. I do not think the decision around radiotherapy for Marjolin’s ulcers is as clear cut as it is for breast cancer as there remains a lack of evidence. I have tried to articulate this clearly in the revised manuscript.

Reviewer 2 Report

This report provides a case of BTM used for oncoplastic reconstruction, which is interesting and novel. It is well written and it deserves to be published with minor revisions.

Page 1, line 26: Please check the order of references.

Page 1, line 33: The abbreviation for “split-thickness skin grafts” appears later in the text and should be indicated on its first occurrence.

Page 2: The planned time point of postoperative radiotherapy can be indicated.

Author Response

Sorry about the delay in reply, I have been away on leave. Thank you for your comments and feedback. I have revised the minor grammatical errors and included the conditions for post-operative radiotherapy along with proposed time frame.

Reviewer 3 Report

MacDiarmid and Butler presented a case of circumferential marjolins ulcer excised and reconstructed with BTM and split-thickness skin graft.

Major points:

1. The case is well presented but the use of BTM in reconstructing complex wounds such as those due to skin cancer has been described in the literature. For example, Li et al 2021 published in Anz J Surg (PMID: 34085755) described a few cases of squamous cell carcinoma and basal cell carcinoma treated with excision and reconstruction with BTM. It would be great if the authors could comment on how their case is different/novel. 

2. Introduction is rather short and should be enhanced. A more detailed discussion of BTM and why the authors decide to report on this case should be included. 

Minor points:

3. Please avoid using abbreviations in figure legends. 

4. Title of reference 5 is bolded for no particular reason.

Author Response

Sorry about the delay in reply, I have been away on leave. Thank you for your comments and feedback. All your points are valid and useful.

I agree that BTM has now been used for a small number of cutaneous malignancies but its application has never been tested in the treatment of a Marjolin’s ulcer or a large circumferential wrapping defect associated with malignancy. I think that in many centres this may have been managed with amputation and the case demonstrates a viable alternative option. Furthermore I think it highlights a way of achieving limb salvage without the expense and complication associated with free flap repair which is important in developing countries where there is a higher incidence of the disease.  

I have expanded the introduction as per your request and amended the other 2 minor points.

Round 2

Reviewer 3 Report

The authors have adequately addressed my comments and significantly improved their manuscript. I have no additional comments and recommend publication of this manuscript.